# Local Liver Irradiation Concurrently Versus Sequentially with Cabozantinib on the Pharmacokinetics and Biodistribution in Rats

**DOI:** 10.3390/ijms24065849

**Published:** 2023-03-19

**Authors:** Yu-Chuen Huang, Pei-Ying Hsieh, Li-Ying Wang, Tung-Hu Tsai, Yu-Jen Chen, Chen-Hsi Hsieh

**Affiliations:** 1Department of Medical Research, China Medical University Hospital, Taichung 404, Taiwanchenmdphd@gmail.com (Y.-J.C.); 2School of Chinese Medicine, China Medical University, Taichung 404, Taiwan; 3Department of Oncology and Hematology, Far Eastern Memorial Hospital, New Taipei City 220, Taiwan; 4School and Graduate Institute of Physical Therapy, College of Medicine, National Taiwan University, Taipei 100, Taiwan; 5Physical Therapy Center, National Taiwan University Hospital, Taipei 100, Taiwan; 6Institute of Traditional Medicine, College of Medicine, National Yang Ming Chiao Tung University, Taipei 112, Taiwan; thtsai@nycu.edu.tw; 7Department of Radiation Oncology, Mackay Memorial Hospital, Taipei 104, Taiwan; 8Department of Artificial Intelligence and Medical Application, MacKay Junior College of Medicine, Nursing, and Management, Taipei 112, Taiwan; 9School of Medicine, National Yang Ming Chiao Tung University, Taipei 112, Taiwan; 10Division of Radiation Oncology, Department of Radiology, Far Eastern Memorial Hospital, New Taipei City 220, Taiwan

**Keywords:** biodistribution, cabozantinib, hepatocellular carcinoma, pharmacokinetics, radiotherapy

## Abstract

The aim of this study was to evaluate the radiotherapy (RT)-pharmacokinetics (PK) effect of cabozantinib in concurrent or sequential regimens with external beam radiotherapy (EBRT) or stereotactic body radiation therapy (SBRT). Concurrent and sequential regimens involving RT and cabozantinib were designed. The RT–drug interactions of cabozantinib under RT were confirmed in a free-moving rat model. The drugs were separated on an Agilent ZORBAX SB-phenyl column with a mobile phase consisting of 10 mM potassium dihydrogen phosphate (KH_2_PO_4_)–methanol solution (27:73, *v*/*v*) for cabozantinib. There were no statistically significant differences in the concentration versus time curve of cabozantinib (AUC_cabozantinib_) between the control group and the RT_2Gy×3 f’x_ and RT_9Gy×3 f’x_ groups in the concurrent and the sequential regimens. However, compared to those in the control group, the T_max_, T_1/2_ and MRT decreased by 72.8% (*p* = 0.04), 49.0% (*p* = 0.04) and 48.5% (*p* = 0.04) with RT_2Gy×3 f’x_ in the concurrent regimen, respectively. Additionally, the T_1/2_ and MRT decreased by 58.8% (*p* = 0.01) and 57.8% (*p* = 0.01) in the concurrent RT_9Gy×3 f’x_ group when compared with the control group, respectively. The biodistribution of cabozantinib in the heart increased by 271.4% (*p* = 0.04) and 120.0% (*p* = 0.04) with RT_2Gy×3 f’x_ in the concurrent and sequential regimens compared to the concurrent regimen, respectively. Additionally, the biodistribution of cabozantinib in the heart increased by 107.1% (*p* = 0.01) with the RT_9Gy×3 f’x_ sequential regimen. Compared to the RT_9Gy×3 f’x_ concurrent regimen, the RT_9Gy×3 f’x_ sequential regimen increased the biodistribution of cabozantinib in the heart (81.3%, *p* = 0.02), liver (110.5%, *p* = 0.02), lung (125%, *p* = 0.004) and kidneys (87.5%, *p* = 0.048). No cabozantinib was detected in the brain in any of the groups. The AUC of cabozantinib is not modulated by irradiation and is not affected by treatment strategies. However, the biodistribution of cabozantinib in the heart is modulated by off-target irradiation and SBRT doses simultaneously. The impact of the biodistribution of cabozantinib with RT_9Gy×3 f’x_ is more significant with the sequential regimen than with the concurrent regimen.

## 1. Introduction

Hepatocellular carcinoma (HCC) was the sixth most commonly diagnosed cancer and the third leading cause of cancer death worldwide in 2020 [1]. The systemic treatment options available for most cases are limited [2,3,4]; therefore, additional treatment options are needed.

Sorafenib inhibits HCC through the inhibition of vascular endothelial growth factor (VEGF) and partly through the inhibition of the RAS/RAF/MEK/ERK mitogen-activated protein kinase (MAPK) at the level of RAF [5,6]. In the clinic, sorafenib provides a significant improvement in the overall survival of HCC patients [2,3]. Lenvatinib, a multitarget tyrosine kinase inhibitor (TKI), is noninferior to sorafenib [7]. It targets VEGFR1, 2 and 3; fibroblast growth factor receptors (FGFR) 1, 2, 3 and 4; platelet-derived growth factor receptor (PDGFR)-alpha (α); the RET proto-oncogene; and c-kit [8,9]. Cabozantinib (Exelixis, Inc., South San Francisco, CA, USA), as the second-line treatment for HCC, has been reported to result in longer overall survival and progression-free survival than placebo in the CELESTIAL trial [10]. Cabozantinib targets mesenchymal-epithelial transition factor (MET); VEGFR1, 2 and 3; AXL; FLT-3; MER; ROS1; TIE-2; TRKB; TYRO3; RET proto-oncogene; and c-kit [11].

External beam radiotherapy (EBRT) with conventional techniques or stereotactic body radiation therapy (SBRT) is an option for patients with unresectable or medically inoperable HCC [12]. MET and AXL activate the phosphatidylinositol 3 kinase (PI3K)/protein kinase B (AKT) and mitogen-activated protein kinase (MAPK) networks [13,14]. Moreover, cabozantinib targets MET and the TAM family of receptor kinases [11,15]. RT stimulates the transient activation of nuclear factor kappa B (NF-κB) [16,17], and NF-κB increases the expression of PDGF and VEGF [18]. Additionally, the expression of cytochrome P450 3A4 (CYP3A4) and *P*-*glycoprotein* (P-gp) could be affected by RT [19]. Recently, Dawson et al. [20] reported that SBRT followed by sorafenib improved the overall survival, progression-free survival and time to disease progression compared with sorafenib alone for HCC patient. Moreover, our previous data showed that RT modulated PK of TKIs [19,21,22] and supported the interaction between RT and TKIs.

Notably, cabozantinib is a substrate for CYP3A4 [23] and belongs to the TKI group. These data suggest that there may be interactions between RT and cabozantinib. In the current study, the RT–drug interaction of cabozantinib with different RT doses and time schedules was evaluated. Furthermore, the biodistribution of cabozantinib with or without RT was evaluated to provide suggestions for clinical applications.

## 2. Results

### 2.1. Method of Validation for Linearity, Recovery, Precision, Accuracy and Stability

In the current study, the LOD of cabozantinib in the plasma was 0.05 μg/mL. The regression equation for cabozantinib in rat plasma was y = 1.3748x − 0.0599 (r^2^ = 0.9998) (Figure 1). The intraday precision (% RSD) and accuracy (% bias) values were within ± 15%, which were considered to be in the acceptable experimental concentration range. This result indicated that the method was considered acceptable and reproducible. The recovery rate of cabozantinib for 0.05 to 10 µg/mL ranged from 94.2% to 106.7%.

### 2.2. Neither RT_2Gy_ nor RT_9Gy_ Modulated the AUC of Cabozantinib in the Plasma of Freely Moving Rats

There were no statistically significant differences in the concentration versus time curve of cabozantinib (AUC_cabozantinib_) between the control group, RT_2Gy×3 f’x_ and RT_9Gy×3 f’x_ in the concurrent or sequential regimens. However, compared to the control group, the T_max_, T_1/2_ and MRT decreased by 72.8% (*p* = 0.04), 49.0% (*p* = 0.04) and 48.5% (*p* = 0.04) with RT_2Gy×3 f’x_ in the concurrent regimen, respectively. Additionally, the T_1/2_ and MRT decreased by 69.2% (*p* = 0.02) and 66.1% (*p* = 0.02) with RT_2Gy×3 f’x_ in the sequential regimen, respectively. Interestingly, the CL increased by 249.0% (*p* < 0.001) with RT_2Gy×3 f’x_ in the sequential regimen. Compared to RT_2Gy×3 f’x_ given concurrently or sequentially with cabozantinib, the T_max_ increased by 404.1% (*p* < 0.001) with the RT_2Gy×3 f’x_ sequential regimen when compared with the concurrent regimen (Figure 2A).

The T_1/2_ and MRT decreased by 58.8% (*p* = 0.01) and 57.8% (*p* = 0.01) in the RT_9Gy×3 f’x_ concurrent group compared with the control group, respectively. There were no significant differences between the RT_9Gy×3 f’x_ concurrent and sequential regimens with cabozantinib (Figure 2B, Table 1).

### 2.3. Organ Distribution According to Different RT and Lenvatinib Regimens

The biodistribution of cabozantinib in the heart was increased by 271.4% (*p* = 0.04) and 120.0% (*p* = 0.04) with RT_2Gy×3 f’x_ in the concurrent and sequential regimens compared to the concurrent regimen, respectively. Compared to the control group, sequential RT_2Gy×3 f’x_ increased the kidney biodistribution by 31.8% (*p* = 0.02). However, concurrent and sequential regimens in RT_2Gy×3 f’x_ did not affect the biodistribution in the liver, spleen, lung and brain. Interestingly, the biodistribution of cabozantinib was increased by the RT_9Gy×3 f’x_ sequential regimen in the heart and lung by 107.1% (*p* = 0.01) and 44% (*p* = 0.02), respectively, but was not affected by the RT_9Gy×3 f’x_ concurrent regimen when compared with the control group. When compared to the RT_9Gy×3 f’x_ concurrent regimen, the RT_9Gy×3 f’x_ sequential regimen increased the biodistribution of cabozantinib in the heart (81.3%, *p* = 0.02), liver (110.5%, *p* = 0.02), lung (125%, *p* = 0.004) and kidneys (87.5%, *p* = 0.048). No cabozantinib was detected in the brain in any of the groups (Figure 3 and Table 2).

## 3. Discussion

To our knowledge, our study is the first to explore the interaction between RT and cabozantinib. RT did not modulate the AUC of cabozantinib with off-target and SBRT doses. However, RT given concurrently or sequentially with different doses impacted the biodistribution of cabozantinib.

According to the Sorafenib HCC Assessment Randomized Protocol (SHARP) [2] and Asia-Pacific [3] trials and the REFLECT trial [7], sorafenib and lenvatinib are defined as the first-line treatments for patients with HCC. After sorafenib failure, positive outcomes were reported by the RESORCE trial, which confirmed the role of regorafenib as second-line therapy [4]. Additionally, the phase 3 CELESTIAL trial, a study of cabozantinib for HCC, confirmed the favorable outcomes for patients with unresectable HCC who received one or two prior lines of treatment, including sorafenib [10].

Oral TKIs, such as sorafenib, activate VEGFR-2 and VEGFR-3, fms-like tyrosine kinase (FLT)-3, KIT, PDGFRb, RAF, BRAF WT, and BRAF V600E [5,6]. However, the mechanisms of resistance to sorafenib have been elucidated as hypoxia-induced factors, overproduction of VEGF, and inhibition of the RAF/MEK/ERK pathway resulting in the activation of EGFR, AKT/TOR, and mesenchymal-epithelial transition proto-oncogene (c-MET)/hepatocyte growth factor axes [24]. Therefore, overcoming these critical points is crucial in developing effective therapies. Accordingly, regorafenib blocks the activity of protein kinases involving VEGFR1-3, TIE-2, c-kit, Raf-1, c-Ret, V600E-mutated B-Raf, PDGFR and fibroblast growth factor receptor (FGFR) [25,26]. Lenvatinib targets VEGFR1-3, FGFR1-4, PDGFR-α, the RET proto-oncogene and c-kit [8,9]. In contrast to other multi–tyrosine kinase inhibitors, cabozantinib targets VEGFR-1–3 as well as MET and the tumor-associated macrophage (TAM) family (TYRO3, AXL, and MER) of receptor kinases [11,15]. These are induced by hypoxia, and MET and AXL are involved in resistance to antiangiogenic therapy [27]. Additionally, the MET/HGF pathway is involved in HCC progression by promoting cellular proliferation, survival and invasion [28]. In other words, cabozantinib inhibits the development and growth of HCC and resistance to antiangiogenic therapy and may potentially promote an immune permissive environment.

SBRT has been reported as a safe and effective option for high-risk HCC patients unsuitable for or refractory to standard local treatment options according to long-term observation [29]. Recently, Dawson et al. [20] reported that patients with advanced HCC treated by SBRT plus sorafenib had improved overall survival, progression-free survival and time to disease progression compared with those of patients treated with sorafenib alone. The data add to the body of evidence for the role of RT combined with TKIs in patients with locally advanced HCC. Interestingly, our previous data showed that SBRT with sorafenib resulted in a 3-fold higher complete recanalization rate than conventional RT (28% vs. 8%, *p* = 0.014) [19]. Additionally, when considering off-target and target effects, the AUC of sorafenib was increased 4-fold and 1.6-fold in the concurrent RT_9Gy_ groups and the sequential RT_9Gy_ group, respectively [19]. Similarly, the AUC of regorafenib was increased 1.3-fold in the sequential RT_9Gyx3f’x_ group [21]. The AUC of lenvatinib was increased 2-fold in the sequential RT_9Gyx3f’x_ group [22]. The lines of evidence support the interaction between RT and TKIs.

Activation of the PI3K/AKT and MAPK pathways is a well-known trait in cancer. Compounds or modalities that inhibit signaling pathways provide an attractive approach to strengthening the effectiveness of antitumor therapy. RT exposure activates the expression of the MAPK and PI3K pathways [30]. MET and AXL activate the PI3K/AKT and MAPK networks [13,14]. Cabozantinib targets MET and the TAM family of receptor kinases [11,15]. Recently, S49076, a MET inhibitor, has been reported to improve the efficacy of radiotherapy [31]. Moreover, in an in vitro study, the combination of RT with cabozantinib for triple-negative 4T1 cells inhibited clonogenic survival, and a synergistic effect was found [32]. These lines of evidence provide the rationale for the combination of cabozantinib and RT.

In the current study, compared to the control group, the T_1/2_ and MRT decreased in both the RT_2Gy×3 f’x_ concurrent and sequential regimens and RT_9Gy×3 f’x_ concurrent group, respectively. Moreover, the CL increased with RT_2Gy×3 f’x_ in the sequential regimen. However, the current data supported that there were no significant interactions between the RT doses and cabozantinib in the AUC_cabozantinib_ or the RT regimens and cabozantinib in the AUC_cabozantinib_. According to the report by Abou-Alfa GK et al. [10], the rate of grade 3 or 4 adverse events (AEs) of cabozantinib for HCC treatment was approximately 10–17%, and the serious adverse events rate was 12%. Interestingly, in patients with metastatic renal cell carcinoma concurrently treated with cabozantinib and conventional RT or SBRT, the grade 3–4 AE rates were 6.3% and 3.6% of patients and did not increase by adding RT, respectively [33]. Additionally, an in vivo study reported that tumor growth control was not increased by the combination of cabozantinib and irradiation [32]. The study reinforces the concept that there may be no significant interaction between RT and cabozantinib.

RT causes the production of reactive oxygen species that further activate Toll-like receptor 4 (TLR4) followed by the NF-κB-based inflammatory pathway [34]. Nevertheless, cabozantinib inhibits the inflammatory response and apoptosis by inhibiting the activation of the TLR4/NF-κB and NLRP3 inflammasome pathways [35]. Furthermore, cabozantinib is metabolized by cytochrome P450 3A4 (CYP3A4) [23], and the expression of CYP3A4 could be affected by RT [19]. However, cabozantinib has a long half-life of approximately 110 h [36] and high plasma protein binding ability (≥99.7%) [37] that would have required a long washout period (at least 4–5 half-lives) of cabozantinib to wash out the CYP3A modulator’s effect. Additionally, plasma concentration–time profiles show a second absorption peak approximately 24 h after administration, which suggests that cabozantinib may undergo enterohepatic recirculation [38]. Altogether, these data may implicate the possibility of an insignificant interaction between RT and cabozantinib. However, more studies and exploration are needed to confirm this hypothesis.

Numerous preclinical studies have investigated the combination of targeted therapy, chemotherapy, or radiotherapy with immunotherapy to overcome limited responses for the treatment of HCC. RT can enhance antigen presentation by increasing major histocompatibility complex class 1 (MHCI), but responses can be diminished through increased PD-L1 expression, thereby providing a rationale for combination therapy [39]. Additionally, RT can also lead to immunogenic cell death (ICD) through effects that include the release of tumor antigens, exposure of heat shock proteins (HSPs) on the cell surface, release of calreticulin, and increased expression of MHCI, intracellular adhesion molecule 1 (ICAM1), and lymphocyte function–associated antigen 3 [40]. The Gas6/Axl signaling pathway promotes progression, metastasis, immune evasion, and therapeutic resistance in many cancer types [41]. Genetic deletion of AXL resulted in sensitization of tumor cells to radiation and checkpoint immunotherapy [42]. Notably, cabozantinib targets MET and the Tyro3/Axl/Mer family of receptor kinases [11,15], which may promote an immune environment and make tumor cells more sensitive to immune-mediated killing [43,44]. Moreover, the current data provided evidence that there was no significant interaction of PKs between RT and cabozantinib. Accordingly, the combination of RT and cabozantinib decreases the risk of side effects caused by interaction and represents a potential combination therapy for HCC.

According to the phase III study of HCC treated by cabozantinib alone, the rate of grade 3 or 4 AEs was 10–17%, including palmar–plantar erythrodysesthesia (17%), hand-foot syndrome, hypertension (16%), increased aspartate aminotransferase level (12%), fatigue (10%), and diarrhea (10%) [10]. The serious AEs included liver failure, bronchoesophageal fistula, portal vein thrombosis, upper gastrointestinal bleeding, pulmonary embolism, and hepatorenal syndrome [10]. Serious gastrointestinal perforation or fistula has been observed with cabozantinib with 1.2–3%. The risk of grade 3 or above hemorrhagic events caused by cabozantinib is 2.1–3%. Additionally, osteonecrosis of jaw caused by cabozantinib is 1% [38]. Although patients with metastatic renal cell carcinoma are treated with cabozantinib and conventional RT or SBRT, the AEs do not increase when compared with those of cabozantinib [33]. When the liver and renal function are impaired, the AUC of cabozantinib could be increased by 63–81% and 7–30%, respectively [38]. Moreover, it should be noted that cabozantinib is associated with a higher risk of causing cardiovascular damage [45]. Cabozantinib use in patients with metastatic renal cell carcinoma caused significant heart failure [46,47,48]. However, a patient-derived xenograft model of papillary renal cell carcinoma carrying an activating mutation of MET treated by cabozantinib caused striking tumor regression and inhibited lung metastasis [49].

In the current study, the respective biodistribution of cabozantinib in the heart was increased by 271% and 120% with RT_2Gy×3 f’x_ in the concurrent and sequential regimens compared to the concurrent regimen. Moreover, sequential RT_2Gy×3 f’x_ increased kidney biodistribution by 32%. Furthermore, sequential RT_9Gy×3 f’x_ increased the biodistribution of the heart by 107% and increased that of the lung by 44%. When cabozantinib combined with RT, the biodistribution of heart, lung and kidney could be modulated. It increases the chance of shrinking the metastatic tumor in the lung, but the potential risk to the heart should be considered, especially for patients with impairment of liver and renal function.

The cross-talk between RT and c-Met has suggested that c-Met inhibition such as cabozantinib could be used as a strategy to increase cellular radiosensitivity [50]. Accordingly, during combination therapy, the functions of the heart, lung, kidney and liver still need to be considered with caution because RT can modulate the biodistribution of cabozantinib at different doses and regimens. In particular, SBRT delivers highly consistent dose distributions and extends a larger volume of low-to-moderate doses around the target than three-dimensional RT [51]. In addition, the SBRT sequential regimen increased the biodistribution of cabozantinib in the heart, liver, lung and kidneys compared with the concurrent regimen. Additionally, in future combination strategies and the design of clinical trials, these unexpected biological enhancements of cabozantinib with RT should be addressed cautiously to avoid severe toxicity when RT and cabozantinib are used as collaborative tools in treatment strategies.

Several limitations should be considered in the interpretation of the present study. First, the current study used a free-moving SD rat model but not orthotopic or heterotopic models to explore the interaction between RT and cabozantinib. The current data did not support the interaction between RT and cabozantinib in rats. Coincidently, in an in vivo study reported by Reppingen, N. et al. [32], tumor growth control was not increased by cabozantinib plus irradiation. Moreover, the AEs did not increase by adding RT with cabozantinib for patients with metastatic renal cell carcinoma [33]. Therefore, using a nontumor model to evaluate the interaction between RT and cabozantinib appears reasonable. Second, the current study did not include a disease model treated with cabozantinib and RT to evaluate the treatment effects of the combination of RT and cabozantinib. However, the current analysis sheds light on the unexpected biodistribution caused by the combination. Additionally, no interaction between RT and cabozantinib was noted. All of these are useful for prospective clinical trial designs. Third, the presence or absence of an interaction between cabozantinib and RT could not be ensured before the study; therefore, the current study did not explore the possible mechanism. However, the lines of evidence provide the rationale for radiosensitization with cabozantinib for RT [11,15,30,32,52,53]. Further studies to detect the possible mechanism in vivo and in vitro are warranted in the future.

## 4. Materials and Methods

### 4.1. Chemicals and Reagents

Cabozantinib was purchased from Sigma‒Aldrich (St. Louis, MO, USA). Biochanin A, as an internal standard, was purchased from Toronto Research Chemicals Inc. (North York, ON, Canada). Polyethylene glycol 400 (PEG 400) and heparin sodium were purchased from Sigma-Aldrich. Pentobarbital sodium was obtained from SCI Pharmtech (Taoyuan, Taiwan). The solvents and reagents for chromatography were purchased from J.T. Baker (Phillipsburg, NJ, USA) and Merck (Darmstadt, Germany). Standard solutions of lenvatinib and biochanin A were stored in methanol at −20 °C. Triply deionized water from Millipore (Bedford, MA, USA) was used for all preparations.

### 4.2. High-Performance Liquid Chromatography-Ultraviolet (HPLC-UV)

The HPLC system consisted of a chromatographic pump (LC-20AT), an online injector (SIL-20C) equipped with a 10 μL sample loop to inject the sample and an ultraviolet detector (SPD-M20A). Cabozantinib and samples were separated on an Agilent ZORBAX SB-phenyl column (150 × 4.6 mm i.d. particle size = 5 μm). The mobile phase for the cabozantinib group was water and acetonitrile (30:70, *v*:*v*) at a flow rate of 1 mL/min. The optimal photodiode-array detection for cabozantinib was set at a wavelength of 251 nm. The retention time of cabozantinib was 6.9 min with good separation and no endogenous interference in the rat plasma samples, and the procedure exhibited good selectivity (Figure 4). Biochanin A was used as the internal standard (IS), and the retention time of biochanin A was 3.8 min (Figure 4).

### 4.3. Method Validation: Calibration Curve

The calibration curves covered a concentration range from 0.05 to 10 μg/mL. The linearity of the assay was determined using the coefficient of determination (*𝑟*^2^) for the calibration curve, which should be greater than 0.995. The limit of detection (LOD) was the concentration that generated a signal-to-noise ratio of 3, and the lower limit of quantification (LLOQ) was the lowest concentration of the linear regression that yielded a signal-to-noise ratio of 10. The 0.05 μg/mL limit of quantification was the lowest concentration on the calibration curve that could be measured routinely with acceptable bias and relative SD. Calibration standards of plasma samples were prepared by adding known amounts of cabozantinib (10 μL) into blank rat plasma (40 μL) to yield a range of 0.05–10 μg/mL. These mixtures were supplemented with 150 μL of internal standard solution (1 μg/mL).

### 4.4. Method validation: Precision, Accuracy and Recovery

The intra-assay variabilities for cabozantinib were determined by quantitating six replicates at concentrations of 0.05, 0.1, 0.5, 1, 5 and 10 μg/mL using the HPLC method described above on the same day (intraday) and for six consecutive days (interday). The accuracy (% bias) was calculated as follows: accuracy (% bias) = [the nominal concentration (*C*_nom_) − the mean value of observed concentrations (*C*_obs_)/*C*_nom_] × 100. The relative standard deviation (RSD) was calculated as follows: precision (% RSD) = [standard deviation (SD)/*C*_obs_] × 100. The same data were used to determine both accuracy and precision. The intraday precision (% RSD) and accuracy (% bias) values were within ± 15%, which were considered to be in the acceptable experimental concentration range. Recovery was assessed by comparing the peak areas of the spiked samples post extraction with the standard solution at 0.05, 0.5 and 10 μg/mL.

### 4.5. Experimental Animals and Drug Administration

#### 4.5.1. Animals and Sample Preparation

The study protocol was reviewed and approved by the Institutional Animal Experimentation Committee of National Yang Ming Chiao Tung University, Taipei, Taiwan, and by the Institutional Animal Care and Use Committee (IACUC, approval number 1100322). Adult male Sprague–Dawley rats (300 ± 20 g body weight) were provided by the Laboratory Animal Center at National Yang Ming Chiao Tung University (Taipei, Taiwan). All animal experiments followed the guidelines and procedures for the care of laboratory animals at National Yang Ming Chiao Tung University.

#### 4.5.2. Irradiation Technique

A freely moving rat model was designed for the current study [19]. Briefly, the whole liver for the EBRT technique or central area 1.5 × 1.5 cm in size for the SBRT technique of rats was localized by computed tomography. For the whole liver field, the cranial margin was set 5 mm from the top of the diaphragm, and the caudal margin was set 5 mm lower than the liver margin. The whole liver was targeted for irradiation. The experimental animals were randomized to groups receiving sham RT, RT 2 Gy (RT_2Gy_) and RT 9 Gy (RT_9Gy_) with three fractions with concurrent or sequential cabozantinib. The data were collected from six rats per group.

#### 4.5.3. Drug Delivery with RT under Different Time Schedules and Doses

A radiation dose of 2 Gy was considered the daily conventional dose or the off-target dose around the target that received an ablation RT dose. Nine Gy was used to simulate the SBRT dose. The animals were divided into five groups as follows: (A) sham group, cabozantinib (6 mg/kg) only for 3 days (d) with RT_0Gy_ (cabozatinib_×3 d_); (B) whole liver (2 Gy with 3 fractions) RT_2Gy×3 f’x_ concurrent with cabozantinib (6 mg/kg) for 3 days; (C) whole liver RT_2Gy×3 f’x_ sequential with cabozantinib (6 mg/kg) for 3 days; (D) SBRT (9 Gy with 3 fractions) RT_9Gy×3 f’x_ concurrent with cabozantinib (6 mg/kg) for 3 days; and (E) RT_9Gy×3 f’x_ sequential with cabozantinib (6 mg/kg) for 3 days (Figure 5). Rats were initially anesthetized with pentobarbital (50 mg/kg, i.p.) and remained anesthetized throughout the experimental period. After the surgery, the rats were placed in an experimental cage and allowed to recover for 1 day. Cabozatinib was dissolved in triply deionized water and administered (6 mg/kg, p.o.) to the rats (n = 6 per group).

#### 4.5.4. Sample Preparation

Blood samples were collected via polyethylene tubing (PE-50) implanted into the jugular vein of each rat in a heparin-rinsed vial. An aliquot of 100–120 µL of blood was collected at time intervals of 0, 5, 15, 30, 45, 60, 90, 120, 150, 180, 210 and 240 min following drug administration. At each time point, 200 μL of blood was drawn into heparin-rinsed Eppendorf tubes and then centrifuged at 13,000 rpm for 10 min at 4 °C to obtain plasma. Plasma was stored at −20 °C until analysis. Each collected blood sample was transferred to a heparinized microcentrifuge tube and centrifuged at 13,000 rpm for 10 min. The resulting plasma (50 μL) was then mixed with 150 μL of internal standard solution (10 μg/mL). The denatured protein precipitate was separated by vortexing for 20 s and centrifuged at 13,000 rpm for 10 min at 4 °C.

#### 4.5.5. Organ Distribution

Six hours after cabozantinib administration for 3 days (6 mg/kg, p.o.), blood samples were collected as mentioned above. The brain, liver, heart, spleen, lung and kidney were collected and weighed. These collected samples were stored at −20 °C until analysis.

#### 4.5.6. Organ Samples

The thawed organ samples were homogenized in 50% aqueous acetonitrile (the ratio of sample weight to volume was 1:5), and the homogenate was centrifuged at 13,000× *g* for 10 min at 4 °C. The supernatant was collected, placed in brown Eppendorf tubes, and stored at −20 °C until analysis. Briefly, each organ sample (50 μL) was combined with 150 μL of IS solution (diethylstilbestrol) for protein precipitation. Finally, 20 μL of filtrate was subjected to HPLC analysis.

### 4.6. Pharmacokinetics and Data Analysis

Pharmacokinetic parameters, including the area under the concentration versus time curve (AUC), clearance (CL), elimination half-life (*𝑡*_1/2_), volume of distribution at steady state (Vss) and mean residence time (MRT), were calculated using the pharmacokinetics calculation software WinNonlin Standard Edition, Version 1.1 (Scientific Consulting, Apex, NC, USA) by a compartmental method. Relative bioavailability (RB %) = (AUC_irradiated_/AUC_control_) × 100. The results are presented as the means ± standard deviations.

### 4.7. Calculations and Data Analysis

All statistical calculations were performed with Statistical Product and Service Solutions (SPSS) for Windows, version 20.0 (SPSS, IBM, USA). All data are expressed as the mean ± standard deviation (SD). One-way analysis of variance (ANOVA) was used for comparisons between groups, and statistically significant differences were defined as * 𝑝 < 0.05 or ** *p* < 0.01.

## 5. Conclusions

Taken together, these data suggest that the PK of cabozantinib cannot be modulated by irradiation. However, the biodistribution of cabozantinib can be affected by RT, especially in the heart and kidney. The RT_9Gyx3f’x_ sequential regimen had a greater impact on the biodistribution of cabozantinib than the concurrent regimen. The current understanding of the systemic effects of cabozantinib with localized irradiation could be beneficial for exploring RT as a synergistic tool in HCC treatment strategies.

## Figures and Tables

**Figure 1 ijms-24-05849-f001:**
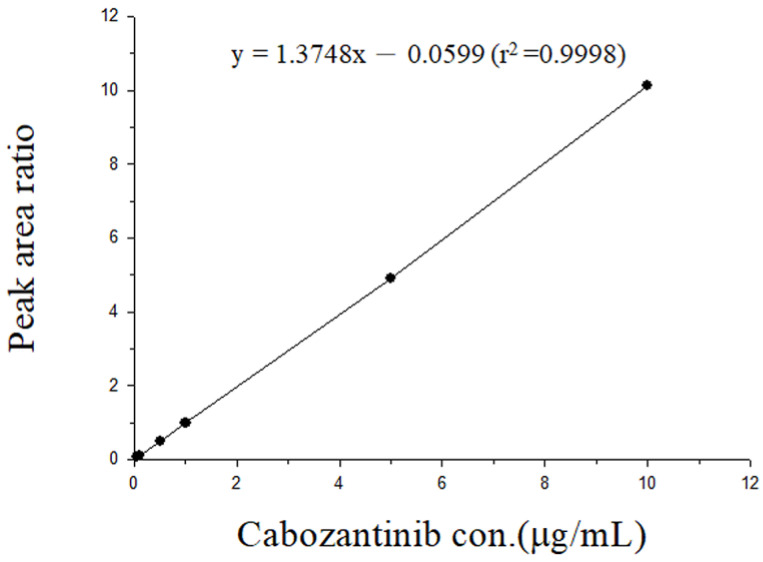
Calibration curve for cabozantinib in the range of 0.05–10 µg/mL.

**Figure 2 ijms-24-05849-f002:**
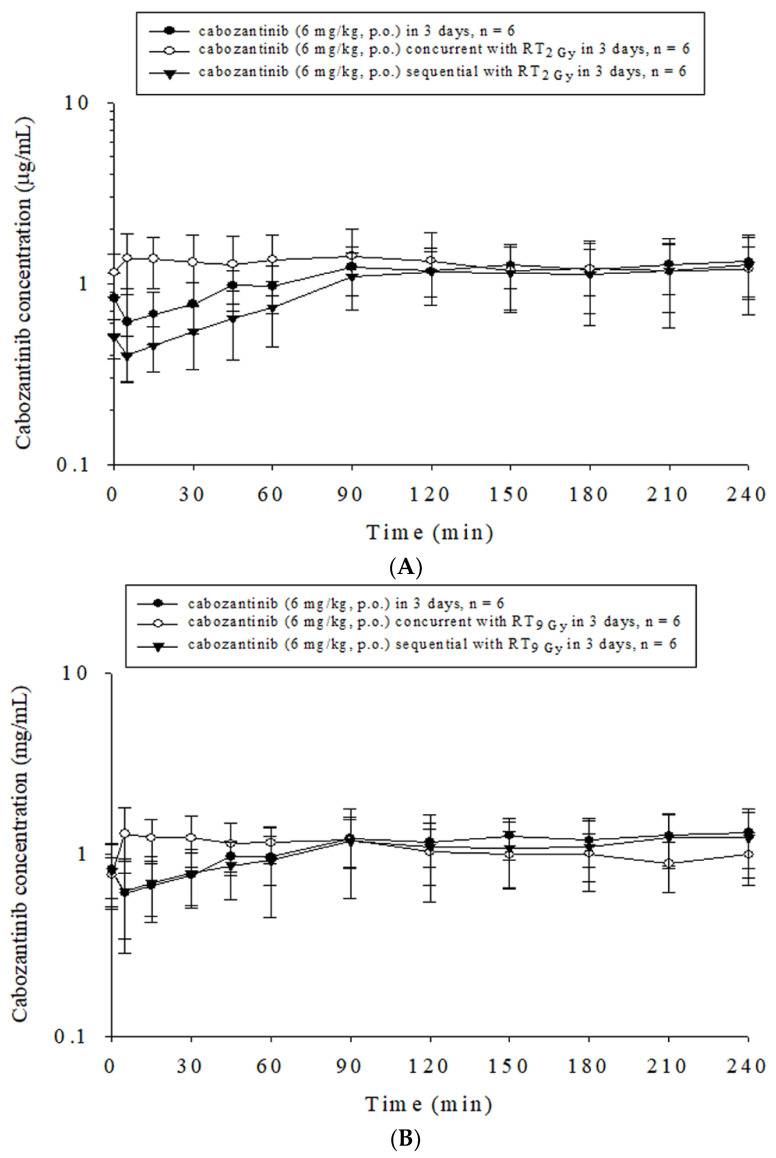
Concentration versus time curves of cabozantinib in the plasma of rats under different time courses with or without radiotherapy (RT). The treated groups included (**A**) the sham group, cabozantinib (p.o., q.d. × 3 d) with RT_0Gy_ (cabozantinib_×3 d_); the concurrent group, cabozantinib_×3 d_ 1 h after RT_2Gy_ in 3 fractions (RT_2Gy×3 f’x_); the sequential group, cabozantinib_×3 d_ 24 h after RT_2Gy×3 f’x_; (**B**) the sham group, cabozantinib_×3 d_; the concurrent group, cabozantinib_×3 d_ 1 h after RT_9Gy_ in 3 fractions (RT_9Gy×3 f’x_); and the sequential group, cabozantinib_×3 d_ 24 h after RT_9Gy×3 f’x_. Data are expressed as the mean ± S.D. (n = 6 per group).

**Figure 3 ijms-24-05849-f003:**
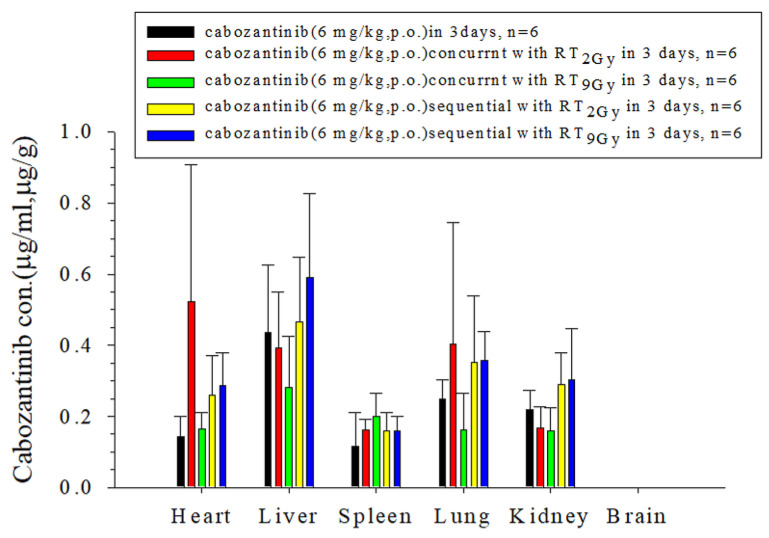
The concentration of cabozantinib in the heart, liver, spleen, lung, kidney and brain of rats after the administration of cabozantinib (6 mg/kg p.o., q.d.) with or without radiotherapy. The cabozantinib concentration units in the organs are expressed as µg/mL or µg/g (n = 6 per group).

**Figure 4 ijms-24-05849-f004:**
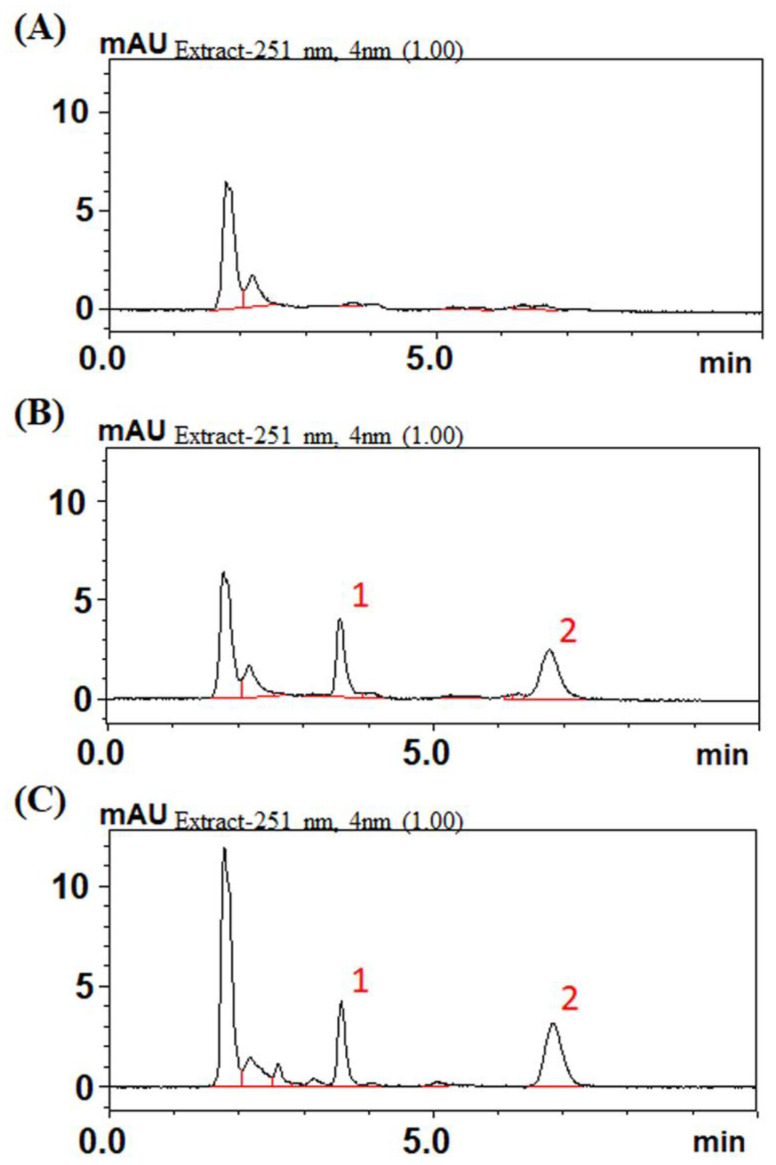
Chromatograms resulting from high-performance liquid chromatography for cabozantinib. (**A**) Blank. (**B**) Blank spiked with cabozantinib (1 µg/mL) and biochanin A (0.75 µg/mL). (**C**) Plasma scheme 60 min after cabozantinib administration (6 mg/kg, p.o.). Peak 1: biochanin A (0.75 µg/mL). Peak 2: cabozantinib (0.8 µg/mL).

**Figure 5 ijms-24-05849-f005:**
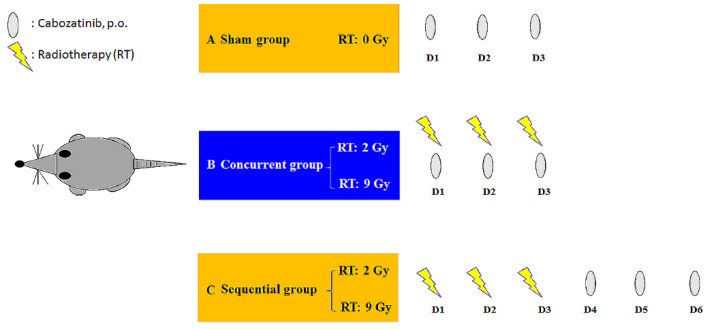
Oral cabozantinib (6 mg/kg p.o., q.d.) with radiotherapy (RT) under different time schedules and RT doses. The rats were divided into (**A**) a sham group, cabozantinib (p.o., q.d. × 3 d) with RT_0Gy_ (cabozantinib_×3d_); (**B**) a concurrent group, cabozantinib_×3d_ 1 h after RT_2Gy_ in 3 fractions (RT_2Gy×3f’x_) and RT_9Gy×3f’x_; and (**C**) a sequential group, cabozantinib_×3d_ 24 h after RT_2Gy×3f’x_ and RT_9Gy×3f’x_. Data are expressed as the mean ± S.D. (n = 6 per group).

**Table 1 ijms-24-05849-t001:** Pharmacokinetic parameters of cabozantinib in rats after administration for 3 days (6 mg/kg p.o., q.d.) with or without radiotherapy (RT, 2 Gy and 9 Gy).

			RT _2Gy_		RT _9Gy_	
PK Parameters	Unit	Cabozantinib 6 mg/kg (n = 6)	RT Concurrent with Cabozantinib 6 mg/kg (n = 6)	RT Sequential with Cabozantinib 6 mg/kg (n = 6)	RT Concurrent with Cabozantinib 6 mg/kg (n = 6)	RT Sequential with Cabozantinib 6 mg/kg (n = 6)
AUC_0-T_	min*µg/mL	266.6 ± 70.6	306.2 ± 113	233.9 ± 92.5	257.6 ± 76.0	250.9 ± 101
T_max_	min	135.0 ± 94.4	36.67 ± 31.7 *	185.0 ± 55.0 ^a^	38.33 ± 70.1	175.0 ± 66.9
C_max_	µg/mL	1.444 ± 0.44	1.599 ± 0.47	1.359 ± 0.54	1.372 ± 0.47	1.353 ± 0.49
T_1/2_	min	1091 ± 362	555.4 ± 351 *	336.5 ± 149 *	449.0 ± 272 *	1578 ± 1196
Vss	mL/kg	4831 ± 1307	3942 ± 1303	5668 ± 2220	4204 ± 1272	3153 ± 482
Cl	mL/min/kg	3.401 ± 1.54	6.695 ± 5.65	11.87 ± 0.74 **	8.766 ± 5.30	2.056 ± 1.77
MRT	min	1601 ± 517	824.4 ± 507 *	543.3 ± 195 *	674.6 ± 386 *	2312 ± 1736

Data are expressed as the mean ± S.D. (n = 6). * *p* < 0.05, compared with cabozantinib only. ** *p* < 0.01, compared with cabozantinib only. a *p* < 0.01, compared with RT 2 Gy concurrent with cabozantinib.

**Table 2 ijms-24-05849-t002:** Concentrations of cabozantinib in the heart, liver, spleen, lung, kidney and brain of rats after administration (6 mg/kg, p.o.) with or without radiotherapy.

		RT _2Gy_		RT _9Gy_	
Organ	Cabozantinib 6 mg/kg (n = 6)	RT Concurrent with Cabozantinib 6 mg/kg (n = 6)	RT Sequential with Cabozantinib 6 mg/kg (n = 6)	RT Concurrent with Cabozantinib 6 mg/kg (n = 6)	RT Sequential with Cabozantinib 6 mg/kg (n = 6)
Heart	0.14 ± 0.06	0.52 ± 0.39 *	0.26 ± 0.11 *	0.16 ± 0.05	0.29 ± 0.09 *^b^
Liver	0.44 ± 0.19	0.39 ± 0.16	0.47 ± 0.18	0.28 ± 0.14	0.59 ± 0.24 ^b^
Spleen	0.12 ± 0.09	0.16 ± 0.03	0.16 ± 0.05	0.20 ± 0.07	0.16 ± 0.04
Lung	0.25 ± 0.05	0.40 ± 0.34	0.35 ± 0.19	0.16 ± 0.10	0.36 ± 0.08 *^b^
Kidney	0.22 ± 0.06	0.17 ± 0.06	0.29 ± 0.09 ^a^	0.16 ± 0.07	0.30 ± 0.14 ^b^
Brain	0.00 ± 0.00	0.00 ± 0.00	0.00 ± 0.00	0.00 ± 0.00	0.00 ± 0.00

Data are expressed as the mean ± SD (n = 6). *: compared with cabozantinib 6 mg/kg × 3 d. *: *p* ˂ 0.05. 1. a: 2 Gy concurrent with cabozantinib 6 mg/kg compared with 2 Gy sequential with cabozantinib 6 mg/kg. a: *p* ˂ 0.05. b: 9 Gy concurrent with cabozantinib 6 mg/kg compared with 9 Gy sequential with cabozantinib 6 mg/kg b: *p* ˂ 0.05..

## Data Availability

Data is contained within the article.

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
