# Peer review of "Local Liver Irradiation Concurrently Versus Sequentially with Cabozantinib on the Pharmacokinetics and Biodistribution in Rats"

_ijms, 2023, doi:10.3390/ijms24065849_

Round 1

Reviewer 1 Report

The manuscript involves the study of pharmacokinetics and biodistribution of cabozantinib in rats following Local liver irradiation concurrently versus sequentially. My main concern is the method of assay and its validation that may dramatically affect the results of the study.

1.       Lines 107-115: In the methodology, the authors mentioned two different ranges for calibration curves. Please explain.

2.       Lines 124-126: (The relative error and coefficient of variation were maintained within ± 15%, except for the LLOQ, which was not permitted to exceed ± 20%)…Does this match the presented results? Please clarify.

3.       Lines 202-205: the following concerns about the method validity should be addressed in details:

·         The author reported only LOD is 0.5 µg/mL, while according to the calibration curve LOQ is 0.05 µg/mL? The authors should clarify this confusion.

·         The % bias and RSD are relatively high.  The authors should revise the acceptable limits for % bias and % RSD according to the USP. These limits should be stated with appropriated cited reference and the reported ranges should conform with such limits.

Reviewer 2 Report

The manuscript “Local liver irradiation concurrently versus sequentially with cabozantinib on the pharmacokinetics and biodistribution in rats” found the change in biodistribution of cabozantinib that could be considered in the clinical practice. This work could be accepted after some minor revisions.

1. The introduction needs a major revision to highlight the rationale of study.

2. Since the biodistribution of cabozantinib was changed, the histological investigation should be added to observe the corresponding change.

Round 2

Reviewer 1 Report

The authors addressed the comments in an acceptable way. the paper can be accepted for publication.